# Cryo-EM structures reveal the H⁺/citrate symport mechanism of *Drosophila* INDY

Subin Kim[1], Jun Gyou Park[1], Seung Hun Choi[1], Ji Won Kim[2], Mi Sun Jin[1]

*Drosophila* I'm Not Dead Yet (INDY) functions as a transporter for citrate, a key metabolite in the citric acid cycle, across the plasma membrane. Partial deficiency of INDY extends lifespan, akin to the effects of caloric restriction. In this work, we use cryo-electron microscopy to determine structures of INDY in the presence and absence of citrate and in complex with the well-known inhibitor 4,4′-diisothiocyano-2,2′-disulfonic acid stilbene (DIDS) at resolutions ranging from 2.7 to 3.6 Å. Together with functional data obtained in vitro, the INDY structures reveal the H⁺/citrate co-transport mechanism, in which aromatic residue F119 serves as a one-gate element. They also provide insight into how protein–lipid interactions at the dimerization interface affect the stability and function of the transporter, and how DIDS disrupts the transport cycle.

## Introduction

The I'm Not Dead Yet (INDY) protein in *Drosophila melanogaster* is a homodimeric plasma membrane transporter composed of 590 amino acids, belonging to the family of divalent anion–sodium symporters (DASS) (Inoue et al, 2002a; Knauf et al, 2002; Knauf et al, 2006; Lu, 2019). The INDY gene is predominantly expressed in metabolic tissues, including the midgut, fat body, and liver-like oenocytes. INDY recognizes intermediates from the tricarboxylic acid cycle as substrates, with a preference for the dicarboxylate form of citrate at pH 6, and the anion exchanger inhibitor, DIDS (4,4′-diisothiocyano-2,2′-disulfonic acid stilbene), strongly inhibits its transport activity (Knauf et al, 2002). Partial loss of INDY expression or activity leads to a significant increase in longevity similar to that observed on caloric-restriction diets (Rogina et al, 2000; Wang et al, 2009; Rogina & Helfand, 2013; Zhu et al, 2014; Costello & Franklin, 2016). Moreover, INDY mutant flies have favorable metabolic profiles, such as reduced lipid accumulation, lower levels of mitochondrial reactive oxygen species and oxidative damage, and preserved intestinal stem cell homeostasis

(Marden et al, 2003; Rogers & Rogina, 2014). The reduced expression of the INDY gene has also been reported to inhibit germ cell proliferation in males with delayed sperm maturation and decreased spermatocyte numbers (Hudry et al, 2019).

Solute carrier gene family 13 member A5 (SLC13A5), also known as NaCT, is the functional counterpart to *Drosophila* INDY in various species, including human (Inoue et al, 2002b; Inoue et al, 2002c), mouse (Inoue et al, 2004), zebrafish (Gopal et al, 2015), and *C. elegans* (Fei et al, 2004). Although their substrate specificities are similar, their transport mechanisms differ significantly. *Drosophila* INDY is a cation-independent, electroneutral dicarboxylate exchanger, whereas human NaCT is a Na⁺-dependent, electrogenic transporter (Inoue et al, 2002a; Inoue et al, 2002c; Knauf et al, 2006). Other notable differences between the transporters of different species involve citrate affinity (Gopal et al, 2007; Gopal et al, 2015), inhibitor selectivity (Pajor et al, 2016; Higuchi et al, 2020), and effects of lithium on transporter function (Inoue et al, 2003; Gopal et al, 2015). These differences have biological consequences. For example, the loss of INDY in *C. elegans* has positive metabolic consequences and extends lifespan, as mentioned above (Rogina et al, 2000; Fei et al, 2004). Similarly, complete loss of NaCT in mice has a range of beneficial metabolic effects, including resistance to obesity, diabetes (Birkenfeld et al, 2011), and non-alcoholic fatty liver disease (Brachs et al, 2016), as well as reduced blood pressure (Willmes et al, 2021). However, loss of NaCT in mice also comes with defects in tooth and bone mineralization (Irizarry et al, 2017). In human, loss-of-function mutations in NaCT have a markedly different outcome, namely, the autosomal recessive disorder Early Infantile Epileptic Encephalopathy-25 (EIEE-25), which is characterized by neonatal epilepsy and severe developmental delay (Thevenon et al, 2014; Bhutia et al, 2017); in addition, uptake of circulating citrate into hepatocytes is reduced, thereby stimulating glycolysis and suppressing the production of fatty acids and cholesterol. These effects are closely related to defects in metabolic disorders such as type 2 diabetes, obesity, and metabolic syndrome (Birkenfeld et al, 2011), and they highlight the complexity of the metabolic pathways affected. Understanding the precise molecular mechanisms underlying the functional consequences of mutations of INDY and its homologs may provide valuable insights

[1]School of Life Sciences, Gwangju Institute of Science and Technology (GIST), Gwangju, Republic of Korea   [2]Department of Life Sciences, Pohang University of Science and Technology (POSTECH), Pohang, Republic of Korea

Correspondence: misunjin@gist.ac.kr

into the roles of these enzymes in metabolism and aging across species.

Here, we report seven cryo-EM structures of full-length INDY in the presence and absence of citrate or DIDS at resolutions ranging from 2.7 to 3.6 Å. These structures provide a framework for understanding the overall transport cycle of INDY. In addition, they offer a mechanistic explanation for H⁺/substrate symport by INDY and the nature of DIDS inhibition, potentially leading to the development of new therapeutic strategies for various metabolic disorders and age-related conditions.

# Results

### Functional characterization of INDY in proteoliposomes

Previous functional studies of INDY have mainly been carried out in *Xenopus* oocytes (Knauf et al, 2002, 2006). To provide further characterization of INDY in vitro, we reconstituted the purified full-length protein into liposomes composed of a 1:1 ratio (w/w) of *Escherichia coli* polar lipids and egg PC (L-α-phosphatidylcholine) and studied its transport activity under various conditions. As seen with the data at the cellular level (Inoue et al, 2002a; Knauf et al, 2002, 2006), INDY-mediated [$^{14}$C]-citrate uptake into the liposomes was markedly stimulated at pH values below 6.0 (Fig 1A). These findings indicate that under physiological conditions, citrate, which has pK$_a$ values of 3.1, 4.8, and 6.4, is preferentially transported in its divalent rather than its trivalent form (Inoue et al, 2002a; Knauf et al, 2006). Because INDY has been identified as a citrate–dicarboxylate exchanger in oocytes (Knauf et al, 2006), we also compared how the presence of internal succinate affects the uptake of [$^{14}$C]-citrate. Contrary to our expectation, the presence of succinate inside the proteoliposome did not lead to increased accumulation of [$^{14}$C]-citrate (Fig 1A). We speculate that this divergence may stem from microenvironmental differences between the liposome system and cellular-level experiments. We showed further that INDY exhibits significantly higher selectivity for citrate than dicarboxylates such as succinate, malate, glutarate, and α-ketoglutarate, as well as monocarboxylates such as lactate (Figs 1B and S1A–F). Kinetic measurements at 22°C indicated that INDY catalyzes citrate transport with a $K_m$ of 1.6 ± 0.2 μm and a maximal rate of transport of 1.8 ± 0.1 nmol/mg/min (Fig 1C). This value is similar to those of mammalian Na⁺/dicarboxylate co-transporters, which typically exhibit micromolar to submillimolar $K_m$ values (Ganapathy et al, 1988; Ogin & Grassl, 1989; Inoue et al, 2002b). Furthermore, INDY-mediated transport of citrate does not depend on a cation gradient as a driving force (Fig 1D) and is almost completely inhibited in the presence of DIDS, with an IC$_{50}$ value of 12.2 ± 2.5 μm (Figs 1E and S1G).

### Citrate is co-transported with protons

Based on the observed pH-dependent substrate uptake of INDY (Fig 1A) and the evidence that some secondary transporters can evolve to shift between Na⁺- and H⁺-dependent substrate binding (Drew & Boudker, 2024; Reddy et al, 2024 *Preprint*), we hypothesized that INDY co-transports citrate with protons. To test this hypothesis, we used the pH-sensitive fluorophore pyranine to track proton movement during the transport cycle (Parker et al, 2014). Proton-based assays were performed at pH 6 both inside and outside liposomes because these conditions are optimal for citrate transport (Fig 1A). We observed that a pH drop in the lumen, indicated by a decrease in pyranine fluorescence (quenching), occurred only upon the addition of citrate to the external solution (Fig 1F). Such acidification was not observed in the presence of succinate or DIDS (Fig 1F). Valinomycin, a potassium ionophore, did not affect the INDY-mediated proton influx, confirming that INDY is indeed an electroneutral transporter (Inoue et al, 2002a; Knauf et al, 2006). Therefore, our biochemical characterization confirmed that INDY is a DIDS-sensitive H⁺/citrate symporter that predominantly recognizes citrate in its divalent form.

### Overall structures of apo INDY

Apo-state structures of INDY were determined under two conditions: using nanodiscs at pH 6 and 8 (Borhani et al, 1997; Denisov et al, 2004). In both conditions, INDY had a homodimeric structure in which the topologically inverted protomers contained 11 transmembrane helices (TM 1–6 and TM 8–12), two helix-turn-helix hairpins (HP1 and HP2), and four helices parallel to the membrane (TM 4c, TM 7, TM 10c, and TM13) (Figs 2A and S2). Each protomer was divided into two domains: a central scaffold domain (TM 1–4 and 8–10), which plays an indispensable role in stable dimerization of the transporter, and a peripheral transport domain (HP1, HP2, TM 5–7, and TM 11–13) responsible for citrate binding and transport. The horizontal TM 4c and TM 10c helices serve as connectors between these two functionally independent domains.

The apo structures occurred in three conformations: (i) outward-open and (ii) inward-open conformations in which the substrate-binding site is exposed to the extracellular and intracellular sides, respectively, and (iii) an asymmetric conformation containing elements of both conformations (Fig 2B and C). At pH 6, INDY exhibits two of these conformations, outward-open (43%) and asymmetric (57%) (Figs S12, S13A–F, S14, S15A–F, and S16). Conversely, at pH 8, it adopts the inward-open (32%) and asymmetric (68%) conformations, whereas the outward-open state is not sampled at all (Figs S17, S18A–F, S19, S20A–F, and S21). Superposition of the individual protomers did not indicate any major structural differences as a function of pH, as indicated by a Cα r.m.s.d. < 1.8 Å (Fig S3A and B). Rather, pH change seems to cause a marked shift in population distribution, specifically the absence of the outward-open conformational state at pH 8.

The ability of INDY to adopt multiple configurations in the apo state highlights the intrinsic dynamics of the transporter independent of substrate binding (Fig 2B and C). Furthermore, the existence of an asymmetric conformation suggests that the two transport domains of INDY can slide independently across the membrane via an elevator-type mechanism (Garaeva & Slotboom, 2020). In contrast, the dimerization interface between the two scaffold domains, which is estimated to have a buried surface area of ~2,600 Å (Krissinel & Henrick, 2007; Krissinel, 2015), is primarily stabilized by extensive van der Waals and hydrophobic interactions (Fig S4A) so that the scaffold framework remains relatively rigid during the conformational transitions (Fig S4B).

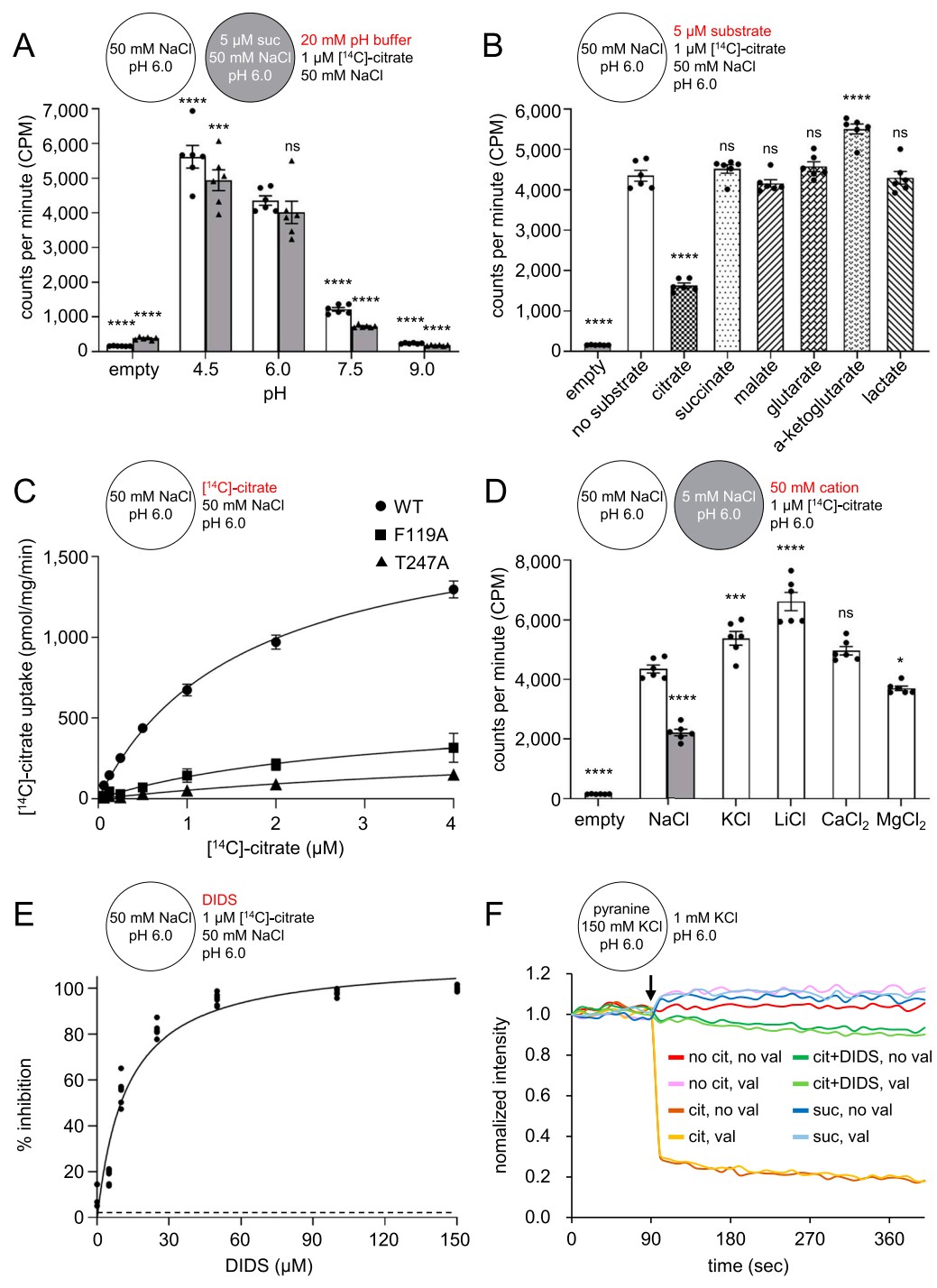

**Figure 1. In vitro transport activity of INDY.**
**(A)** pH-dependent [$^{14}$C]-citrate transport activity of INDY in liposomes. Uptake of 1 μm external [$^{14}$C]-citrate was measured for 5 min in the presence (gray bars) or absence (white bars) of internal unlabeled 5 μm succinate. The external pH of the liposomes was varied from 4.5 to 9.0, whereas the inside pH was maintained constant at pH 6. Empty liposomes at external pH 6 were used as a negative control. **(B)** Substrate specificity of INDY. Uptake of external 1 μm [$^{14}$C]-citrate in the presence of 5 μm potential substrates was measured in buffers at a pH of 6. **(C)** Kinetic analysis of citrate transport by WT INDY and mutants reconstituted in proteoliposomes. **(D)** Cation-independent [$^{14}$C]-citrate transport activity of INDY in liposomes. Uptake of external 1 μm [$^{14}$C]-citrate in the presence of 50 mM of various cations was measured in buffers with a pH of 6. The internal NaCl concentration was fixed at 50 mM but was lowered to 5 mM when establishing a NaCl gradient across liposomes. **(E)** Dose-dependent inhibition of [$^{14}$C]-citrate transport by DIDS. **(F)** Representative fluorescence traces of the pH indicator pyranine under various conditions. The results demonstrate that proton influx occurs exclusively upon the addition of citrate (black arrow), whereas succinate and a valinomycin-induced membrane potential did not induce lumen acidification. Data were normalized to 1 based on the first reading of each experiment to allow comparison between multiple datasets. All data points and error bars are means ± SEM for six independent measurements (or three for kinetic analyses). Statistical significance was analyzed by one-way ANOVA followed by Dunnett's test

## Protein–lipid interaction at the interface of the scaffold domains

When INDY was embedded in nanodiscs, we observed many non-protein densities on hydrophobic areas of the surface and within clefts between the domains. These densities likely correspond to those of phospholipid molecules or sterols, although definite identification is impossible without quantitative lipid analysis. In particular, we observed reliable EM densities corresponding to phospholipids with two fatty acyl chains within the cleft between the two scaffold domains (Fig 3A). The consistent observation of these densities in all INDY structures suggests that protein–lipid interactions in this region are not state-dependent. Because phosphatidylethanolamine (PE) was the most abundant component of the *E. coli* polar lipids used in the nanodisc reconstitution (Dowhan, 1997), we built the molecule 1-palmitoyl-2-oleoyl-sn-glycero-3-phosphoethanolamine (POPE or 16:0–18:1 PE) as a representative lipid model that fits well in the density. The tails of the modeled POPE are mainly surrounded by hydrophobic residues, whereas the head group interacts with polar residues (Fig 3A and B). To further explore the role of this interfacial lipid in INDY, we created mutants by introducing a bulky phenylalanine side chain at position 43 (G43F) or removing the histidine at position 40 (H40A) to disrupt lipid binding at the scaffold interface. Interestingly, when both mutants were reconstituted into a membrane-like environment, they had significantly reduced transport activity (Figs 3C and S5) and a lower melting temperature ($T_m$ = 56 and 56.5°C, respectively) compared with the WT (60°C) (Fig 3D and E). This suggests that the phospholipid at the scaffold interface plays an important role in transporter function and stability (Drew & Boudker, 2024).

## The citrate-binding site

Upon substrate binding, the transport domain undergoes a rigid-body sliding motion involving a ~13 Å downward shift, a ~19° tilt, and a ~43° rotation relative to the scaffold domain, with a Cα r.m.s.d. of 0.7 Å compared with the outward-open apo state (Figs 4A and S6A–C). Consequently, INDY was mostly in the partially open inward-facing intermediate state (accounting for 81% of input particles), with the substrate-binding site of each protomer slightly exposed toward the cytoplasm (Figs 4B, S22, S23A–F, and S24). In this conformation, we observed an additional electron density corresponding to the bound citrate. However, the underlying density was too weak to accurately define the correct citrate conformer, probably because of binding instability and/or low occupancy. A molecular dynamics (MD) simulation analysis in the lipid bilayers confirmed this hypothesis (Hess et al, 2008). When the citrate was manually docked based on the results of site-directed mutagenesis experiments (Fig 5A; see below) and used as the starting structure for a 100 ns MD simulation at pH 6, one of the bound citrate molecules began to drift away from INDY within the first 28 ns (Fig 5B and Video 1). This result suggests a loose binding of citrate in the binding site, thereby facilitating rapid release for efficient transport

activity. Subsequently, the additional ~4 Å downward shift of the transport domain resulted in the substrate-binding site becoming continuous with the cytoplasm, thus facilitating the release of the bound citrate (Fig 4).

In our structure, the citrate-binding site is enclosed by several residues, including the two conserved Ser-Asn-Thr/Val (SNT/V) motifs on HP1 and HP2 (e.g., N170 and A171 from HP1, and N480 and V481 from HP2), as well as F119 on TM 4b, T245, A246 and T247 on the TM 5a-b loop, and T523 and P524 on the TM 11a-b loop (Fig 5A) (Laskowski & Swindells, 2011). Because of the ambiguity concerning the citrate density, our initial plan was to predict its binding mode using the UCSF DOCK 6 program (Allen et al, 2015), but this was unsuccessful (Fig S6D). That is, the top 5 docked poses of citrate predicted by DOCK 6 do not align with either the cryo-EM electron density map or the results from alanine mutagenesis analyses. We speculated that the failure might be due to our inability to adjust the protonation state of citrate during the simulation (i.e., citrate was simulated in tricarboxylate form). Accordingly, we manually built a tentative model of citrate, placing it at the center of the region defined by the residues mentioned above to maximize favorable local interactions with the protein while avoiding steric collisions. In this hypothetical binding mode, citrate interacted with surrounding residues, forming a network of hydrogen bonds, and ionic and van der Waals interactions (Fig 5C). Furthermore, the substantial positive charge of the binding site likely not only facilitates the recruitment of negatively charged citrate, but also strengthens its binding (Fig 5D).

To confirm the proposed mode of binding of citrate, we performed site-directed mutagenesis experiments and [$^{14}$C]-citrate transport assays with proteoliposomes. Residues S169 and S479 of the SNT/V motif were chosen as background controls because they are close to the substrate but do not participate in binding (Figs 5A and S6E). As anticipated, single alanine mutations introduced in the citrate-binding site were sufficient to lead to almost complete loss of substrate transport activity, whereas mutation of the control S169A had no significant effect (Figs 5E and S5). In contrast, the S479A control mutant exhibited an unexpected increase in transport activity compared with the WT protein. One possible explanation for this enhanced transport could be that the mutation alleviates steric hindrance or alters the local hydrogen bonding network within the binding pocket, thereby facilitating substrate binding and/or translocation. Using mutant T247A, we further evaluated the impact of the mutations on INDY transport kinetics. The T247A mutant had a higher $K_m$ (6.2 μm) and lower $V_{max}$ (0.38 nmol/mg/min) for citrate transport than the WT (1.6 μm and 1.8 nmol/mg/min, respectively), indicating an apparent decrease in transport efficiency (Fig 1C). Nano-differential scanning fluorimetry (nanoDSF) measurements also demonstrated that all mutants had lower melting temperature (Tm) values than the WT and exhibited insignificant or small Tm increases upon the addition of 100 μM citrate (Figs 5F and G and S7A–I). Tm measurement for the N170A mutant could not be performed because of its high thermal instability. Conversely, we observed a marked 7.2°C ± 0.5 increase

comparing other samples to control. ns indicates $P < 0.1234$, * indicates $P < 0.0332$, ** indicates $P < 0.0021$, *** indicates $P < 0.0002$, and **** indicates $P < 0.0001$. Schematics above the histograms in each panel give the internal and external compositions of the proteoliposomes.

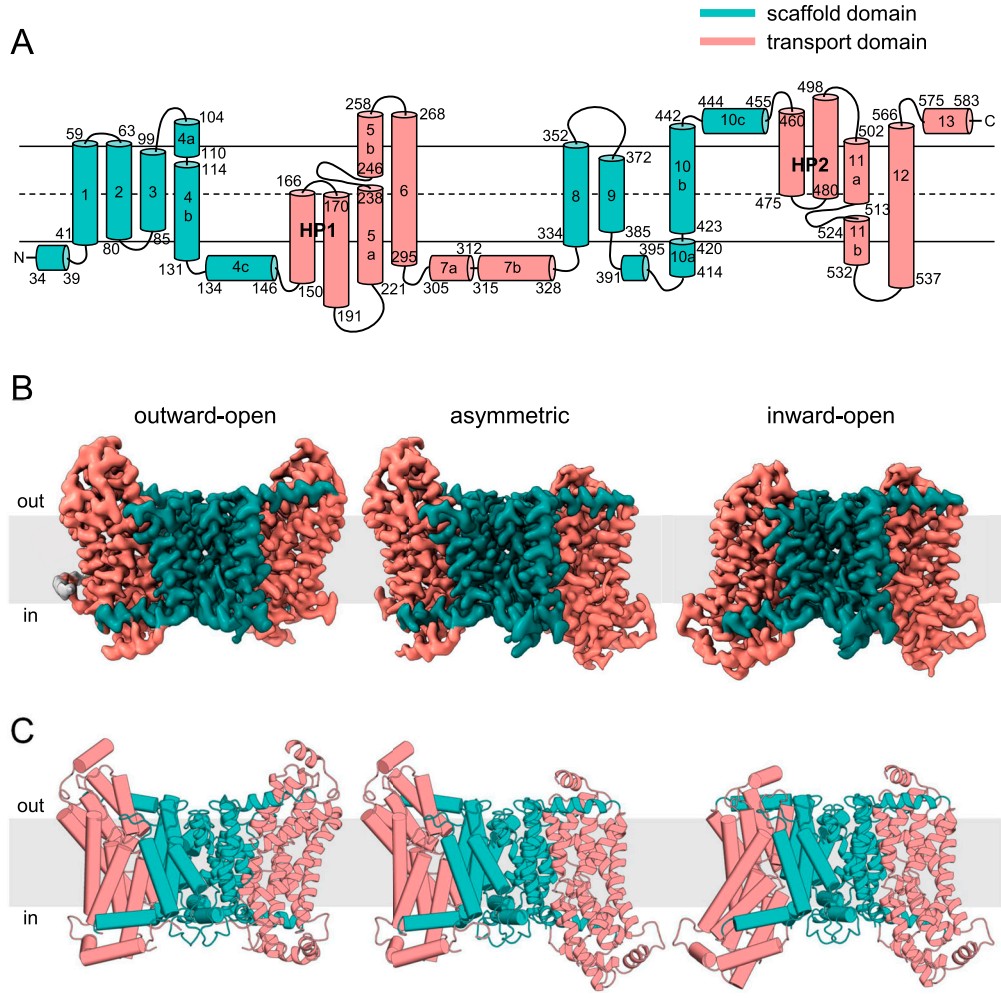

**Figure 2. Three different conformational states of apo INDY.**
**(A)** Schematic representation of the topology of INDY. Scaffold domain and transport domain helices are colored teal and pink, respectively. **(B)** Cryo-EM maps of apo INDY viewed from within the plane of the membrane. **(C)** Overall structures of INDY in the apo state. One protomer is depicted in cylinder representation, the other in cartoon representation.

when adding citrate to the WT protein, suggesting that citrate likely occupies the binding site through interactions with the proposed residues, thereby enhancing thermal stability.

### F119 serves as a gatekeeper controlling the direction of substrate translocation

Another notable feature of the structural transition is a rearrangement of aromatic residue F119 on TM 4a in the scaffold domain (Fig 4A). In the outward-open state, the substrate-binding site is positioned near the extracellular side at the membrane outer leaflet, and F119 adopts an upright orientation with respect to the membrane plane (Fig 4A, left). This configuration allows the substrate to access the binding site from the extracellular side, while preventing its release inward (Fig 4B, left). During substrate binding and the downward movement of the transport domain, the side chain of F119 rotates by ~80° (Fig 4A, middle). As a result, F119 moves to within 3 Å of the citrate (Fig 5A), thereby effectively trapping the bound citrate in the middle of the bilayer (Fig 4B, middle). F119 also appears to be a gatekeeper on the

extracellular side in the inward-open state, blocking the escape of bound substrate to the outside (Fig 4A and B, right). In agreement with these structural analyses, the F119A/Y/L mutants had significantly reduced transport activity (Fig 5E). Specifically, kinetic studies revealed that the F119A mutant had a higher $K_m$ (3.0 $\mu$m) and lower $V_{max}$ (0.5 nmol/mg/min) than the WT (1.6 $\mu$m and 1.8 nmol/mg/min, respectively) (Fig 1C). Furthermore, although the F119A/Y/L mutants exhibited melting temperatures similar to the WT, the presence of citrate resulted in apparent Tm values that were 3°C, 6°C, and 7°C lower, respectively, than the WT (Fig 5F and G). These results indicate that substitution of a bulky phenylalanine residue for a smaller amino acid (F119A/L) abolishes the key role of F119 in stabilizing citrate within the binding site. Although tyrosine is also a bulky residue (F119Y), its hydroxyl group seems to induce steric hindrance affecting nearby residues. Consequently, a slight repositioning of the tyrosine is expected, which likely impedes substrate coordination and the normal transport cycle.

The flipping of F119 also appears to regulate the transbilayer movement of the transport domain. When we modeled the side chain

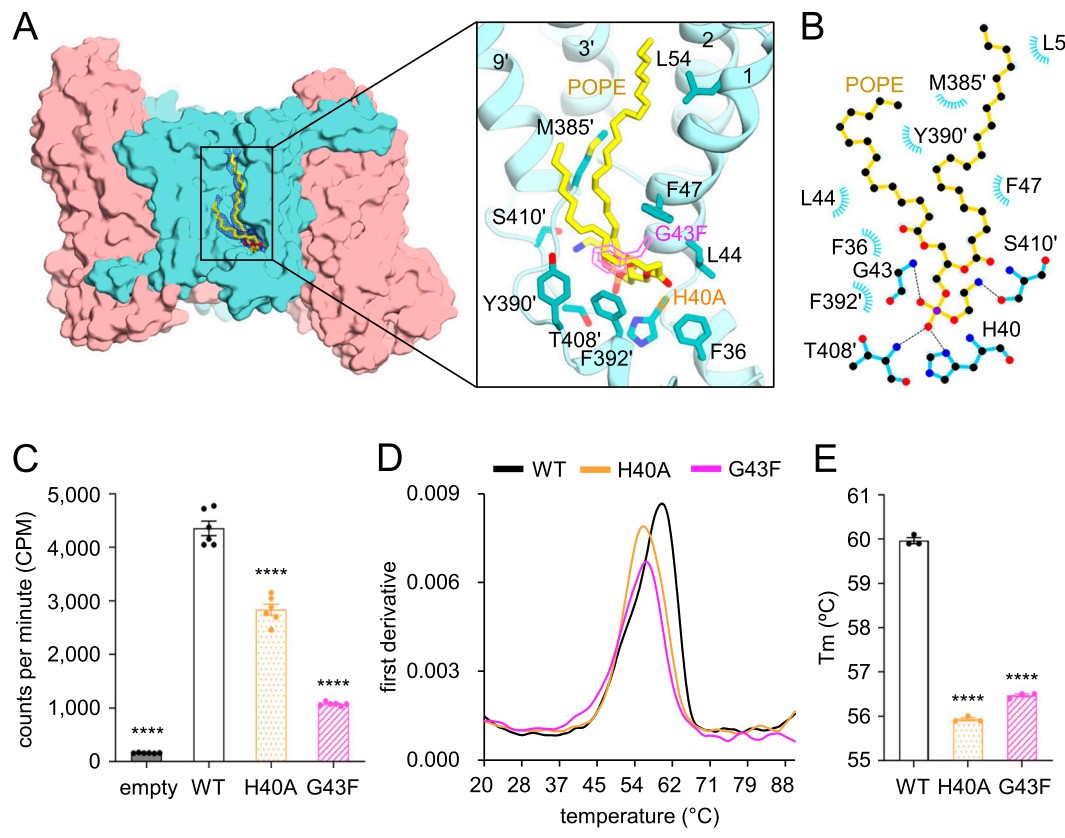

**Figure 3. Lipid-binding site.**
**(A)** Lipid-like density observed at the interface between the scaffold domains in the apo-asymmetric state at pH 8. The tentatively modeled POPE molecule is shown as yellow sticks, and the EM density map (3.5 $\sigma$) is displayed as blue mesh. The lipid-coordinating residues are depicted as sticks. Mutated residues at positions 40 and 43 are highlighted as magenta and orange empty sticks, respectively. **(B)** Interactions between INDY and POPE were analyzed with LigPlot+ software. Residues involved in nonpolar and van der Waals interactions within a distance of 4 Å are depicted as semicircles. **(C)** Comparison of [$^{14}$C]-citrate transport activity of INDY WT and its mutants in proteoliposomes (n = 6). **(D)** Representative thermal unfolding curves showing the thermal stability of WT and mutants in nanodiscs. **(D, E)** $T_m$ values calculated from the curves in panel (D). In (C, E), data points and error bars are means ± SEM of n = 3 independent experiments. Statistical significance was analyzed by one-way ANOVA with Dunnett's test comparing mutants (or empty liposomes) to WT. **** indicates $P < 0.0001$. Single apostrophes are used for all the residues of one protomer to differentiate them from those of the other protomer.

of F119 in an upright position within the citrate-bound structure, we discovered that this state might be energetically unfavorable because of its proximity to L249 on TM 5b when the transport domain tilts toward the scaffold domain (Fig S8A). Moreover, further progression of the transport domain into the inward-open conformation could be hindered by significant steric clashes between these two residues (Fig S8B). In a similar way, in the outward-open state, F119 in the downward position cannot be packed against TM 5b of the transport domain without encountering steric hindrance (Fig S8C). Our structures indicate that the conformational transition of the F119 side chain is possible regardless of substrate binding (Fig 4A), but substrate binding likely lowers the energy barrier for switching of F119 from the up to the down position, accelerating the formation of the inward-open state. Therefore, we propose that the conformational change of F119 is associated with elevator-like movement of the transport domain from one side of the membrane to the other. Overall, our findings suggest that residue F119 not only plays crucial roles in transporting substrate directionally via a one-gate elevator mechanism but also controls the movement of the transport domain across the membrane.

## Mechanism of H⁺/citrate symport in INDY

To date, the atomic structures of six INDY homologs have been determined: human NaCT (Sauer et al, 2021), NaS1 (SLC13A1) (Chi et al, 2024), NaDC1 (SLC13A2), NaDC3 (SLC13A3) (Li et al, 2024), and those of two prokaryotes, *Vibrio cholera* (*Vc*INDY) (Sauer et al, 2022) and *Lactobacillus acidophilus* (*La*INDY) (Sauer et al, 2020). Although they share an overall fold with INDY, with TM helices of comparable lengths and trajectories (Fig S9A–F), there are considerable structural differences between the substrate-binding sites. In particular, when the citrate-bound structures of human NaCT and INDY are superposed, while the HP1 hairpins and TM 11a-b loops occupy similar positions, TM 4b, TM 5a-b loop, and HP2 of INDY are displaced ~2–5 Å toward the bound citrate compared with their counterparts in NaCT (Fig 6A and B). This change renders the INDY substrate-binding site more compact (Fig 6C and D). It also results in a smaller spacing between HP1 and the TM 5a-b loop, as well as between HP2 and the TM 11a-b loop, where Na⁺ ions are known to intercalate and reduce conformational flexibility to facilitate substrate binding in human NaCT (Fig 6B) (Sauer et al, 2021; Chi et al,

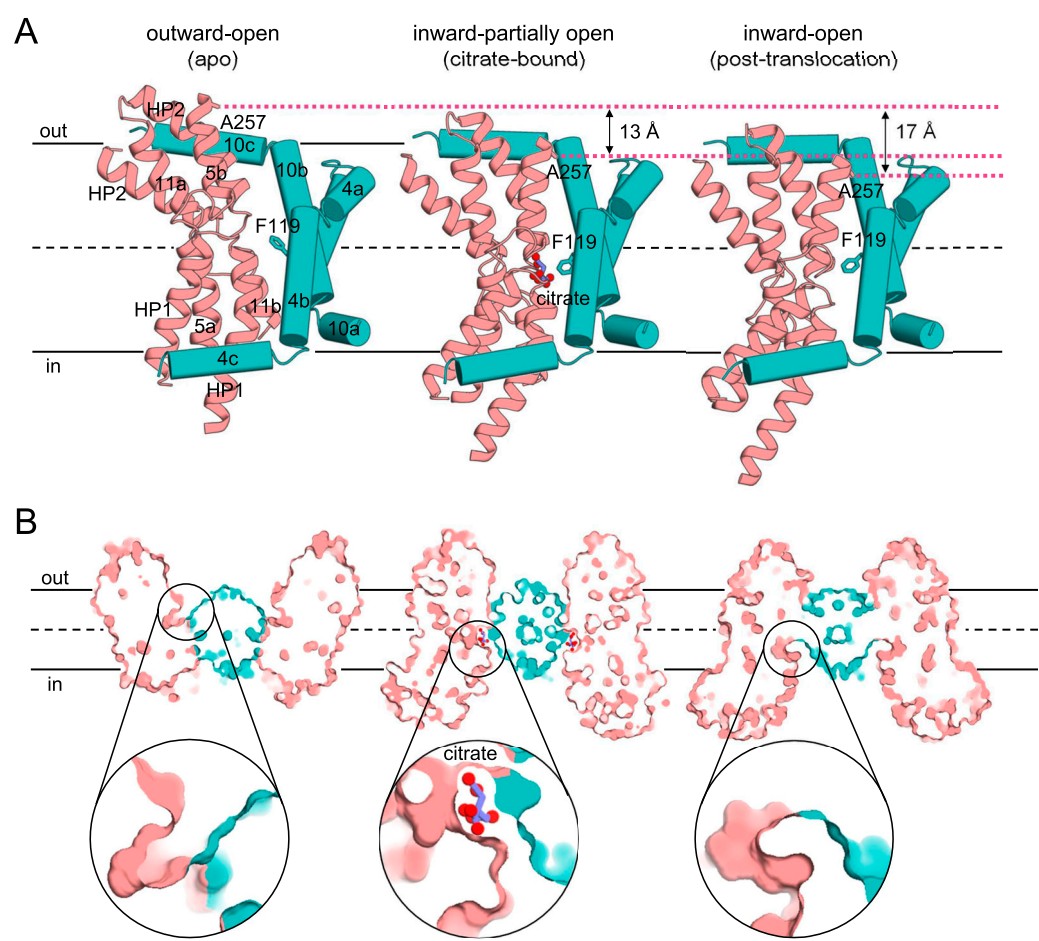

**Figure 4. Elevator-like movement of the transport domain.**
**(A)** Within a protomer, the scaffold and transport domains are illustrated in teal and pink, respectively, for the citrate-free outward-open (left), citrate-bound inward-partially open (middle), and citrate-free inward-open (right) states. The relative positions of the A257 residues and the distances between them in the different conformational states are indicated. Citrate is depicted as a purple ball-and-stick model. The aromatic residue F119 on TM 4b is shown as a stick. **(B)** Surface slab views of the INDY homodimer. A close-up view of the substrate-binding site is shown within the circle.

2024; Li et al, 2024). Consequently, despite the conservation of amino acids involved in Na$^+$ binding (Fig S6E), this spatial restriction likely causes INDY to favor binding of H$^+$ over Na$^+$. As INDY is known to be an electroneutral transporter, at least two protons likely occupy the Na$^+$-binding site to facilitate substrate binding (Inoue et al, 2002a; Knauf et al, 2006).

**Mechanism of inhibition of INDY by DIDS**

Cryo-EM analysis reveals that DIDS-bound INDY exists in two almost equally populated conformations, outward-open and asymmetric (Figs 7A and B, S10A–C, S25, S26A–F, S27, S28A–F, and S29). In contrast to the human NaCT and NaDC3 inhibitors, PF-06649298 (PF2) and PF-06761281 (PF4a), which bind to and stabilize the inward-open transporters (Huard et al, 2015, 2016; Pajor et al, 2016; Sauer et al, 2021; Li et al, 2024), the corresponding electron density for the DIDS molecule is only observed when the protomer is in the outward-open state (Figs 7B and S10C). Structural comparison between the apo and DIDS-bound conformations resulted in a Cα r.m.s.d. of 0.6 Å, with only subtle changes at the substrate-binding site (Fig S10D).

The DIDS density is also broad and ambiguous, confirming its dynamic behavior in solution, as observed in the MD simulation (Fig S11A–F and Video 2 and Video 3). We performed further 3D classification and 3D variability analysis using CryoSPARC software (Punjani et al, 2017; Punjani & Fleet, 2021), but this approach did not significantly improve map quality. As an alternative approach, we built a tentative model of DIDS using the LigandFit wizard from the PHENIX suite (Adams et al, 2010). The result indicated that DIDS was surrounded within 4 Å by residues including K113, T115, R371, F433, K440, and S479 (Fig 7C). Consistent with our structural analysis, alanine substitution of these residues led to noticeable reductions in transport activity (Fig S10E). In particular, residues R371 and F433 of the scaffold domain, and S479 of the SNT/V motif appear to engage in key interactions with DIDS because mutation to alanine resulted in significant reductions in DIDS inhibition efficacy compared with the WT (Fig 7D). Previous studies have demonstrated that DIDS almost completely inhibits both INDY- and human NaCT-mediated substrate transport (Inoue et al, 2002a; Knauf et al, 2002, 2006; Jaramillo-Martinez et al, 2021). Analysis of the sequence conservation pattern indicates that the key residues of

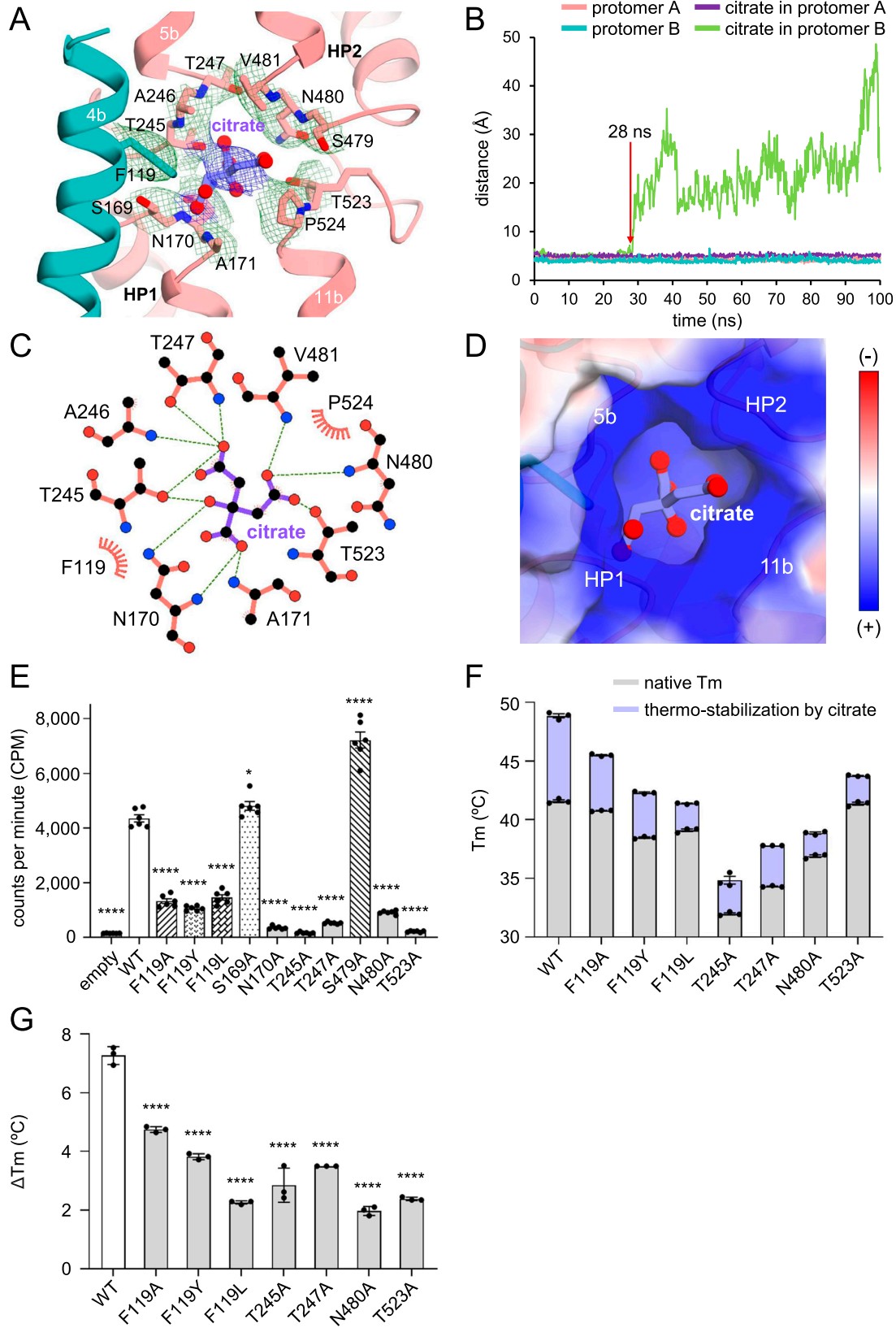

INDY for DIDS binding are well conserved in human NaCT (Fig S10F) (Guex & Peitsch, 1997). Taken together, our findings suggest that DIDS is a competitive inhibitor of citrate binding to INDY. Furthermore, it effectively locks the transporter in the outward-open conformation, impeding the sliding movement of the transport domain from the extracellular space toward the cytoplasm that is crucial for citrate translocation.

# Discussion

In this study, we have examined the biochemical properties of INDY incorporated into liposomes (Fig 1) and presented seven snapshots of the cryo-EM data to reveal the mechanism of substrate translocation and its inhibition (Fig 8). Our findings highlight the atypical nature of *Drosophila* INDY within the DASS family. The most intriguing feature is that although most of the DASS proteins rely on Na$^+$ ions for substrate co-transport, *Drosophila* INDY is Na$^+$-independent, as is *La*INDY, although the bases of their Na$^+$ independence differ (see below) (Inoue et al, 2002a; Knauf et al, 2002, 2006; Sauer et al, 2020). Specifically, structural studies of human NaCT (Sauer et al, 2021) and *Vc*INDY (Nie et al, 2017; Sauer et al, 2022) have shown that the binding of Na$^+$ ions is a prerequisite for substrate binding. The binding sites for two Na$^+$ ions are positioned near the substrate-binding site, enclosed by the flexible loops of HP1 and HP2, and the broken helices corresponding to TM 5b and TM 11b. NMR-style analysis of *Vc*INDY (Sauer et al, 2022) indicates that there may be significant structural plasticity around the substrate-binding site before Na$^+$ binding, preventing it from the binding substrate; however, binding of the Na$^+$ ions could then stabilize the HP loop structures, facilitating stable substrate binding. In contrast, our structural and functional analyses revealed that the substrate-binding site in INDY is more tightly enclosed by the surrounding loops, making it sterically impossible for Na$^+$ ions to bind (Figs 1D and 6). Instead, this configuration likely favors proton binding between the loops (Fig 1F). Given that the current resolution is insufficient to directly locate protons, comprehensive computational studies combined with functional assays are needed in the future to determine the exact H$^+$/citrate coupling stoichiometry, and identify the key residues responsible for proton incorporation.

This characteristic sets *Drosophila* INDY apart from another Na$^+$-independent DASS member, *La*INDY, and other secondary active transporters that do not rely on a Na$^+$ gradient. In *La*INDY, the two positively charged residues, R159 and H392, occupy the Na1 and Na2 sites, respectively, suggesting they may serve as surrogates for Na$^+$ ions (Sauer et al, 2020). The crystal structure of amino acid, polyamine, and organocation transporter (ApcT) (Shaffer et al, 2009) revealed that an amine group of K158 occupied a position equivalent to the Na2 site in the leucine transporter (LeuT) (Yamashita et al, 2005), which is considered the prototype Na$^+$-coupled secondary transporter. This finding suggests that the reversible protonation and deprotonation of K158 plays pivotal roles in triggering conformational changes and substrate transport during the ApcT cycle. Crystallographic and modeling studies of the carnitine transporter CaiT have shown that residue R262 occupies a position equivalent to Na2 in LeuT (Kalayil et al, 2013). In addition, R262 performs an oscillatory movement, mimicking the binding and unbinding of a Na$^+$ ion required for substrate translocation.

Another notable feature of INDY is that the transport domain exhibits elevator-like movements independent of substrate binding (Fig 2). Furthermore, the two protomers move independently across the membrane. However, the possibility of cooperativity between the protomers has not been experimentally investigated. Finally, our INDY structures offer insights into specific protein–lipid interactions that have not been described in detail in homologous transporters. The lipid-like densities are distributed over the entire surface of the scaffold domain, as well as in the clefts between scaffold domains (Fig 3). This indicates that lipids may affect INDY's stability and/or function by interacting with specific structural motifs in the transporter. These interactions may control the elevator dynamics and could be targets for the design of allosteric modulators. Future research should employ computational and theoretical methods, together with experimental approaches, to gain insight into the detailed relationship between INDY and lipids.

# Materials and Methods

### Cloning, expression, and purification

The full-length gene for INDY (GenBank accession number AF509505.1) was obtained from the DNASU Plasmid Repository and cloned into a modified pVL1393 baculovirus transfer vector (BD Biosciences) containing a C-terminal enhanced green fluorescence protein (eGFP) and a decahistidine (10XHIS) tag followed by a thrombin protease site. Site-directed mutations were constructed

**Figure 5. Citrate-binding site.**
**(A)** Close-up view of the putative citrate interaction area. Residues interacting with the tentatively modeled citrate within a 4 Å distance are shown as sticks. Citrate is depicted as a purple ball-and-stick model. The EM density map of the citrate (purple mesh) and nearby residues (green mesh) is shown at the 3.5 σ level. **(B)** MD simulation. The stability of bound citrate during MD simulation was analyzed by monitoring the time evolution of the distance between the protomer and citrate (green and purple). For comparison, the changes in distance between amino acid residues A246 and F433 within each protomer are also shown (teal and pink). Red arrows indicate estimated dissociation times of citrate. See also Video 1. **(C)** Analysis of interactions between INDY and citrate within a distance of 4 Å obtained using LigPlot+ software (Laskowski & Swindells, 2011). Residues engaged in hydrogen bonding interactions are depicted as dashed lines, whereas those involved in hydrophobic interactions within a distance of 4 Å are represented by semicircles. **(D)** Electrostatic surface potential of the citrate-binding site was calculated using the APBS/PDB2PQR software suite (http://www.poissonboltzmann.org), assuming a pH of 6. The electrostatic surface is represented by a contour level from −25 kT/e (negative, red) to +25 kT/e (positive, blue). **(E)** [$^{14}$C]-citrate transport activities of INDY WT and mutants. **(F)** Meting temperature (Tm) of WT INDY and its mutants in the presence and absence of 100 µm citrate. Citrate-induced increases in Tm are highlighted in purple. Representative NanoDSF thermograms are shown in Fig S7. **(G)** Citrate-induced increase of Tm value (ΔTm) derived from NanoDSF experiments. In (E), values are means ± SEM of six independent measurements. In (F), error bars indicate the mean ± SEM of n = 3 independent experiments. In (E, G), statistical significance was analyzed by one-way ANOVA with Dunnett's test comparing mutants with WT. * indicates $P < 0.0332$, and **** indicates $P < 0.0001$.

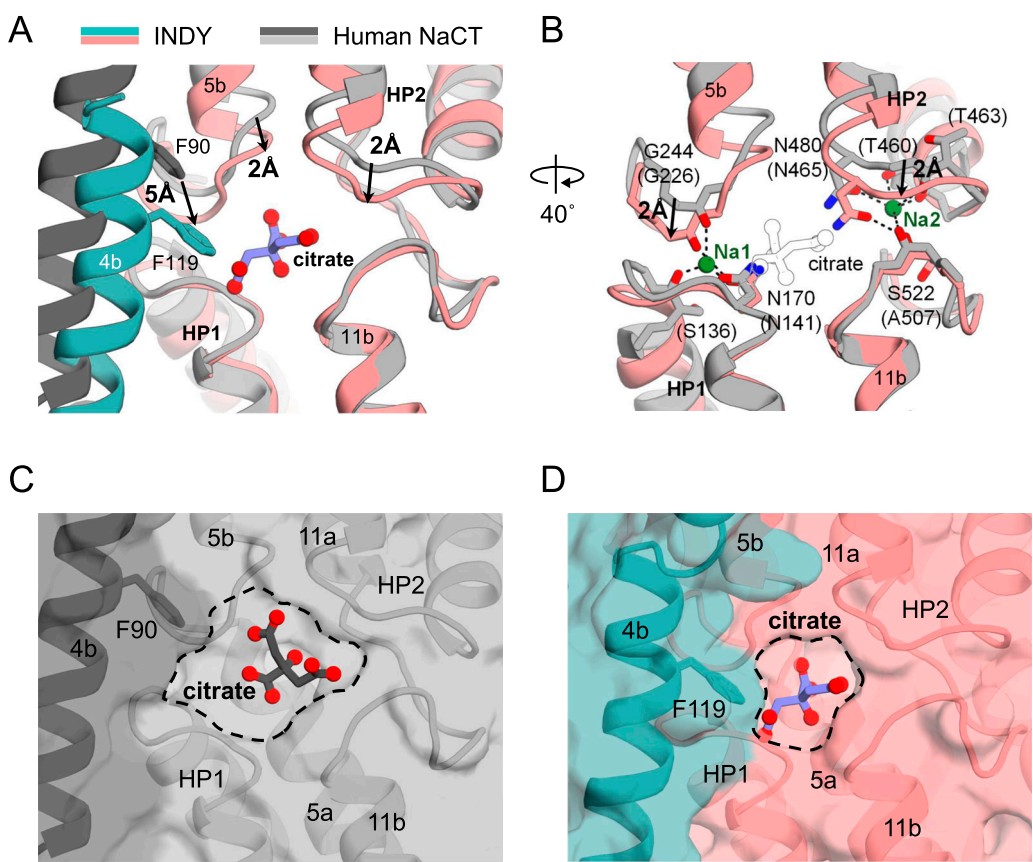

**Figure 6. Comparison of the structures of the citrate-binding sites of INDY and human NaCT.**
**(A, B)** Superpositions of the structures of the citrate (A) and Na⁺ (B) binding sites are shown in ribbon representation for INDY (pink and teal) and human NaCT (PDB ID 7JSK, gray). **(A, B)** Alignment was based on the HP1 region. Citrate is shown as a purple (A) and empty (B) ball-and-stick model, respectively. Sodium ions (Na1, Na2) are depicted as green spheres. Amino acids in parentheses represent substitutions found in human NaCT. The movements of INDY with reference to human NaCT are indicated by arrows. **(C, D)** Surface view of the substrate-binding cavity of human NaCT (C) and INDY (D). The cavities are outlined with dashed lines for guidance.

by overlap PCR and verified by sequencing. Recombinant DNA was transfected into *Spodoptera frugiperda* cells (Sf9, Expression Systems, David CA) using BestBac 2.0 linearized baculovirus DNA (Expression Systems) and Cellfectin II transfection reagent (Gibco). The proteins were overexpressed in *Trichoplusia ni* cells (Hi5, Expression Systems) at 28°C for 72 h post-transfection. Protein expression was monitored by fluorescence microscopy.

After harvesting the transgenic Hi5 cells by centrifugation at 14,000*g* for 10 min, they were resuspended and broken using a microfluidizer in a lysis buffer containing 20 mM Tris–HCl pH 7.5, 300 mM NaCl, 5 mM MgCl₂, 10 μg/ml DNase I (GoldBio), and 0.1 mM PMSF (GoldBio). Cell membranes were collected by centrifugation at 240,000*g* for 1 h and solubilized for 2 h with 2% (wt/vol) n-dodecyl-β-D-maltopyranoside (DDM, Anatrace). Insoluble cell debris was removed by ultracentrifugation at 240,000*g* for 1 h, and the supernatant was loaded onto an anti-GFP DARPin-based affinity resin (Hansen et al, 2017). After thorough washing of the resin with 10 column volumes of lysis buffer containing 0.15% (wt/vol) n-decyl-β-D-maltopyranoside (DM, Anatrace), the bound protein was eluted by overnight thrombin cleavage (Lee Biosolutions). The protein was further purified by size-exclusion chromatography using a Superdex 200 Increase 10/300 GL column (Cytiva) equilibrated with buffer containing 20 mM MES-NaOH pH 6.0, 150 mM NaCl, and 0.15% (wt/vol) DM. All purification steps were performed at 4°C or on ice.

## Nanodisc reconstruction

A vector encoding the C-terminal 6X His-tagged membrane scaffold protein 1 (MSP1), and a deletion mutant lacking the N-terminal 43 residues of human apoA-I (residues 44–243), was prepared as described previously with some modifications (Borhani et al, 1997; Denisov et al, 2004). Briefly, MSP1 was cloned into a pET21a vector (Novagen) and expressed in *E. coli* BL21(DE3) cells cultured in lysogeny broth medium at 37°C. When absorbance at 600 nm (OD₆₀₀) reached 0.6–0.7, protein expression was induced by adding 1 mM isopropyl-β-d-thiogalactopyranoside (IPTG, GoldBio) for 4 h at 30°C. Harvested cells were resuspended and broken using an ultrasonic homogenizer (Branson) in lysis buffer containing 20 mM Tris–HCl pH 8.0, 200 mM NaCl, 10 μg/ml DNase I, and 0.1 mM PMSF. After removal of cell debris by centrifugation at 40,000*g* for 20 min, the supernatant was loaded onto Ni-NTA affinity resin (Incospharm) equilibrated with lysis buffer. The protein was eluted using a buffer containing 20 mM Tris–HCl pH 8.0, 200 mM NaCl, and 300 mM imidazole. To remove the histidine tag, the protein was incubated

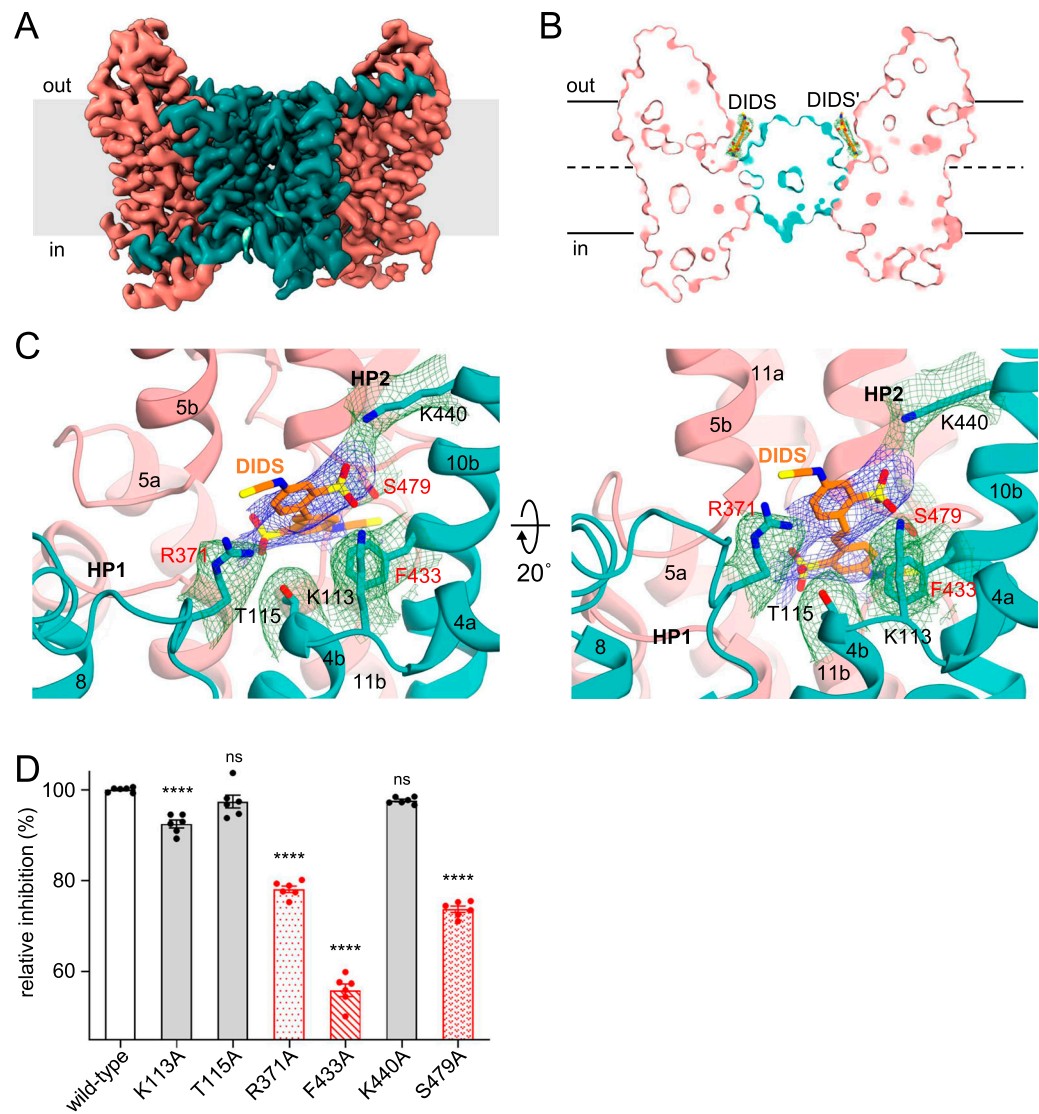

**Figure 7. DIDS inhibition.**
**(A)** Cryo-EM map of DIDS-bound, outward-open INDY viewed from within the plane of the membrane. The scaffold domain and transport domain helices are colored teal and pink, respectively. **(B)** Surface slab view. Orange sticks represent a tentative model of DIDS generated using the PHENIX LigandFit wizard. **(C)** Close-up view of the putative DIDS-binding site. Residues potentially interacting with DIDS, within a distance of 4.0 Å, as analyzed by LigPlot+, are depicted by sticks. **(D)** In particular, key residues (R371, S479, and F433) interacting with DIDS, as validated by the mutagenesis experiments in panel (D), are highlighted in red. The EM density map of the DIDS (blue mesh) and nearby residues (green mesh) is shown at the 3.5 $\sigma$ level. **(D)** Effects of mutations on [$^{14}$C]-citrate transport inhibition by 100 $\mu$m DIDS. Relative inhibition (%) was calculated by normalizing against the WT protein. In (D), all data points and error bars represent means ± SEM of six independent measurements. Statistical significance was analyzed by one-way ANOVA with Dunnett's test comparing mutants to WT. ns indicates $P < 0.1234$, and **** indicates $P < 0.0001$.

with 0.1% (w/w) thrombin protease for 16 h. It was further purified by HiTrap Q anion exchange (Cytiva) and Superdex 200 Increase 10/300 GL gel filtration chromatography.

To initiate nanodisc assembly, the *E. coli* polar lipid extract in chloroform (Avanti Polar Lipids, Inc.) was dried under nitrogen gas. The dried lipid film was rehydrated in buffer containing 20 mM MES-NaOH pH 6.0, and 150 mM NaCl. After water bath sonication, the lipid solution was dissolved in 1% (wt/vol) DDM and mixed with purified MSP1 and INDY at molar ratios of 150:3:1. After incubation at 4°C for 30 min, the remaining detergent was gradually eliminated by treating the sample with three successive changes of Bio-Beads

SM-2 resin (Bio-Rad). Empty nanodisc and excess MSP1 were removed by Superdex 200 Increase 10/300 GL gel filtration chromatography equilibrated with 20 mM MES-NaOH pH 6.0, and 150 mM NaCl. All reconstitution steps were performed on ice or at 4°C.

**Proteoliposome preparation**

*E. coli* polar lipid extract and egg PC in chloroform (Avanti Polar Lipids, Inc.) were mixed in 1:1 molar ratio, and the chloroform was removed with nitrogen gas to create a thin lipid film. The dried lipids were resuspended in reconstitution buffer containing 20 mM

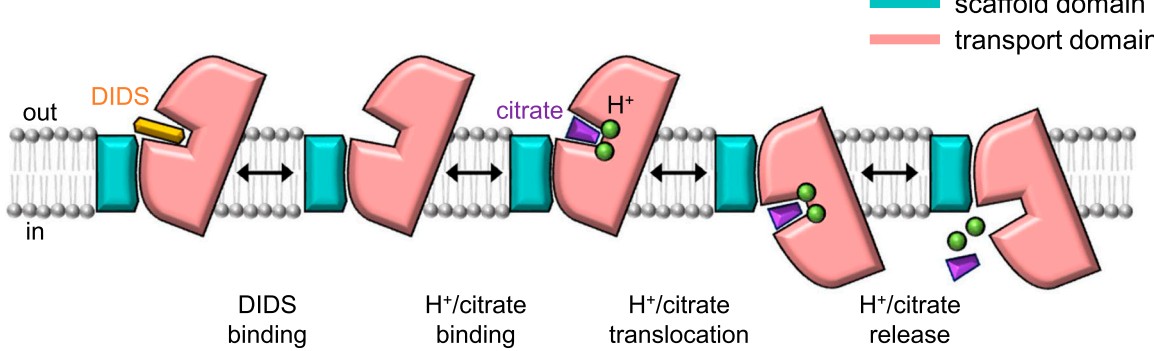

**Figure 8. Proposed transport cycle of INDY and its inhibition by DIDS.**
The schematic diagram depicts a simplified model for the transport cycle of INDY. For simplicity, only one protomer of the INDY dimer is shown, and each protomer is presumed to function independently. Although the exact H⁺/citrate stoichiometry remains to be fully elucidated in future studies, because INDY is known to be an electroneutral transporter, at least two protons are likely co-transported with citrate.

MES-NaOH pH 6.0, and 50 mM NaCl for [$^{14}$C]-citrate transport assay, or 20 mM MES-NaOH pH 6.0, 150 mM KCl, and 1 mM pyranine (trisodium 8-hydroxypyrene-1,3,6-trisulfonate; MedChemExpress) for H⁺ transport assay. The insoluble lipid mass was solubilized by water bath sonication. Lipids were then destabilized by adding 10 mM DDM and 5 mM MgCl₂, resulting in a final lipid concentration of 10 mM. To form unilamellar liposomes, the lipid mixture was extruded using Mini-Extruder equipped with a polycarbonate membrane filter (400 nm pore) (Avanti Polar Lipids, Inc.) on a 40°C heat block. The resulting lipid suspension was mixed with detergent-purified INDY at a ratio of 25:1 (w:w) and gently shaken at 4°C for 2 h. The detergent was subsequently removed by incubating the sample with Bio-Beads SM-2 resin with three exchange steps over a total of 18 h. The proteoliposomes were collected by ultracentrifugation at 165,000$g$ for 2 h at 4°C and resuspended in buffer containing 20 mM MES-NaOH pH 6.0, and 50 mM NaCl for [$^{14}$C]-citrate transport assay, or 20 mM MES-NaOH pH 6.0, and 1 mM KCl for H⁺ transport assay. Final concentrations of proteoliposomes were determined by the phosphorus assay (Itoh et al, 1986). Briefly, 10 µl of proteoliposomes was added to 300 µl of 70% perchloric acid (Sigma-Aldrich) and 10 µl of 2% (wt/vol) ammonium molybdate (Sigma-Aldrich), and the mixture was incubated on a heat block at 150°C for 30 min. The mixture was added to 1.5 ml of 0.4% (wt/vol) ammonium molybdate and 250 µl of 9% (wt/vol) L-ascorbic acid (Sigma-Aldrich) followed by incubation on a heat block at 100°C for 20 min. After cooling to RT, lipid concentration was measured from absorbance at 820 nm using a standard curve prepared with NaH₂PO₄ solution. The proteoliposomes were diluted to a final concentration of 5 mM in the buffer described above and used in [$^{14}$C]-citrate and H⁺ transport assays.

### The [$^{14}$C]-citrate transport assay

To initiate uptake or exchange activity, 67 µl of INDY-loaded proteoliposomes was mixed with 34 µl of reaction buffer containing 20 mM MES-NaOH pH 6.0, 50 mM NaCl, and 1 µM [$^{14}$C]-citrate (PerkinElmer). The mixture was incubated for 5 min at RT, filtered through a 0.22 µm nitrocellulose membrane (Merck Millipore), and washed with 5 ml of deionized water. The washed filter membrane was transferred into a scintillation vial and dissolved in 2 ml of 100% glacial acetic acid (Sigma-Aldrich). Then, 3 ml of liquid scintillation cocktail (Filter-Count, PerkinElmer) was added, and the retained radioactivity of [$^{14}$C]-citrate was determined using a liquid scintillation counter (Hidex 300 SL, HIDEX). For substrate exchange experiments, the internal proteoliposome buffer was supplemented with 5 µm succinate (Sigma-Aldrich). To assess the pH dependence of transport activity of INDY, the proteoliposomes were incubated with reaction buffers at pHs ranging from 4.5 to 9.0. To investigate the substrate specificity of INDY, the proteoliposomes were incubated with reaction buffer supplemented with 5 µm of different substrates. To evaluate the impact of different cations on the transport activity, the proteoliposomes were incubated with reaction buffers containing 50 mM of various cations, such as NaCl, KCl, LiCl, CaCl₂, or MgCl₂. To determine the extent of DIDS (Sigma-Aldrich)-mediated inhibition of the transport activity, the proteoliposomes were incubated with reaction buffer supplemented with varying concentrations of DIDS. To measure the background uptake as a negative control, empty liposomes were incubated in the respective reaction buffer.

### Proton transport assay

H⁺-coupled citrate transport activity was measured using the pH-sensitive fluorescent indicator pyranine. A 50 µl volume of proteoliposomes containing pyranine was diluted with 100 µl of reaction buffer (1 mM KCl) adjusted to pH 6 in a black 96-well plate. Baseline fluorescence was measured using SpectraMax Gemini XPS Microplate Reader (Molecular Devices) at excitation and emission wavelengths of 460 and 510 nm, respectively. Measurements were taken at 10 s intervals over a 90 s period to establish a baseline. To initiate H⁺ transport, 1 mM citrate (or succinate) and 1 µm valinomycin were rapidly added to the reaction mixture. 1 mM DIDS was simultaneously added with citrate to evaluate its inhibitory effect on H⁺ transport. Fluorescence changes were monitored at 10 s intervals for 5 min. A decrease in the fluorescence ratio indicated lumen acidification within liposomes. All measurements were performed at room temperature.

## Nano-differential scanning fluorimetry

The differential scanning fluorimetry method was performed using Prometheus NT.48 (NanoTemper) as previously described (Kotov et al, 2019). Briefly, 10 $\mu m$ of purified protein was incubated with and without 100 $\mu m$ citrate (or DIDS) at room temperature for 10 min. Samples (15 $\mu l$ each) were loaded into standard capillaries (NanoTemper) and subjected to a thermal gradient from 20°C to 90°C at a rate of 2°C per minute. Fluorescence emission from tryptophan after UV excitation at 280 nm was collected at 330 and 350 nm with a dual-UV detector. Melting curves were analyzed with PR.ThermControl software (NanoTemper), wherein the inflection point served as the criterion for determining the melting temperature ($T_m$).

## MD simulations

MD simulation was performed using the GROMACS 2024.1 package (Van Der Spoel et al, 2005) and the CHARMM36m forcefield (Huang et al, 2017). The parameter files for the simulation systems were generated using CHARMM-GUI (Jo et al, 2008; Wu et al, 2014; Lee et al, 2016). The protonation state of each residue was established based on pKa calculations at pH 6. The model membrane was composed of POPC (1-palmitoyl-2-oleoyl-sn-glycero-3-phosphatidylcholine), POPE (1-palmitoyl-2-oleoyl-sn-glycero-3-phosphatidylethanolamine), and POPG (1-Palmitoyl-2-oleoyl-sn-glycero-3-phosphatidylglycerol) with a 2:1:1 stoichiometry for both the upper and lower leaflets. The system in a box of 110 × 110 × 119 $Å^3$ (120 × 120 × 123 $Å^3$) was solvated with TIP3P water molecules (Jorgensen et al, 1983), and the net charge of the system was neutralized with 150 mM NaCl. The resulting simulation system contains a total of 15,579 atoms for the INDY–citrate complex and 44,533 atoms for the INDY-DIDS complex. The particle mesh Ewald method with a cutoff of 14 Å was employed to calculate long-range electrostatic interactions for the INDY–citrate complex (and 19 Å for the INDY-DIDS complex) (Darden et al, 1993, Essmann et al, 1995), and the bond lengths involving hydrogen atoms were constrained using the LINCS algorithm (Hess, 2008, Hess et al, 1997). Before starting the simulations, the systems were minimized with the steepest descent method until the maximum force reached below 1,000 kJ/mol/nm. Simulations were conducted at 303 K using the Nosé–Hoover thermostat (Hoover et al, 1982, Nosé, 1984), and the pressure was maintained at 1 bar employing a Parrinello–Rahman barostat (Parrinello & Rahman, 1981). A time step of 2 fs was used throughout the simulations, and snapshots of the coordinates were analyzed for every 100 ps. Subsequent analyses were performed on the trajectories comprising 2,000 frames obtained from the simulations. All protein images and videos were generated with UCSF Chimera (Pettersen et al, 2004).

## Molecular docking

UCSF Dock 6.10 was used to perform docking studies (Allen et al, 2015). We conducted docking with one protomer of INDY to avoid potential errors and accelerate the simulation. The protein preparation for docking, such as removing water molecules, adding

hydrogens, and assigning charges to polar residues, was performed using the default settings of the Dock Prep tool in UCSF Chimera (Pettersen et al, 2004). The citrate-binding site was determined by "sphgen" estimation with a minimum sphere radius of 1.4 Å. The grid file was generated using a grid box of dimensions 25 Å × 27 Å × 30 Å, with a default grid spacing of 0.3 Å. After the ligand's energy was minimized through minor conformational changes, a flexible docking procedure was followed by the calculation of 1,000 orientations. The poses were viewed using UCSF Chimera's ViewDock tool, and the best-scoring poses were ranked based on the van der Waals energy (vdw_energy) and DOCK 6 grid score. To validate our docking method, control experiments were conducted by employing the same workflow and options in each procedure using the structure of human NaCT containing the inhibitor PF-06649298 (PDB ID 7JSJ) (Sauer et al, 2021). The simulated results showed that the best docking poses with the lowest vdw_energy (−39.9 kcal/mol), rather than the grid score (−49.3), exhibit a better alignment between the predicted and experimental binding orientations. Accordingly, the vdw_energy was used as a criterion to rank the citrate docking results.

## Cryo-EM sample preparation and data collection

Cryo-EM grids were prepared by applying 3 $\mu l$ of protein at a concentration of 0.35–0.7 mg/ml onto freshly glow-discharged Quantifoil R1.2/1.3 or R0.6/1.0 300-mesh Au grids (Quantifoil). For the citrate- or DIDS-bound form, 5 mM or 150 $\mu m$ of the respective compound was added to the protein and incubated on ice for 30 min. Subsequently, the grids were blotted for 3–4 s at a blot force of 1–4 and cryo-cooled by plunging into liquid ethane using Mark IV Vitrobot (Thermo Fisher Scientific) under 100% humidity at 4°C.

Cryo-EM data for apo INDY in nanodiscs at pH 6.0 were collected using a 300 kV Titan Krios cryo-electron microscope (Thermo Fisher Scientific) equipped with a K3 direct detector and BioQuantum energy filter operated at 18 eV slit width (Gatan). A total of 11,663 micrographs were collected at a nominal magnification of X105,000, with a pixel size of 0.826 Å. Each micrograph was dose-fractionated to 50 frames with a total exposure time of 2.9 s. The electron dose rate was 14.9 $e^-$/pix/s (~1.2 $e^-$/$Å^2$/frame), resulting in a total dose of 60 $e^-$/$Å^2$/frame. The defocus range was set to vary from −1.0 to −2.2 $\mu m$ in 0.2 steps.

For the apo INDY in nanodiscs at pH 8.0, a dataset of 11,397 movies was collected using a 300 kV Titan Krios cryo-electron microscope (Thermo Fisher Scientific) equipped with a K3 direct detector and BioQuantum energy filter (Gatan). Micrographs were recorded with EPU software at a nominal magnification of X105,000, corresponding to a pixel size of 0.851 Å. Each movie was dose-fractionated to 50 frames with a total exposure time of 2.9 s. The electron dose rate was 14.9 $e^-$/pix/s (1.2 $e^-$/$Å^2$/frame), resulting in a total dose of 60 $e^-$/$Å^2$/frame. The defocus range was from −0.8 to −2.2 $\mu m$.

For the citrate-bound form in nanodiscs, data were collected from two EM grids (5,589 and 6,585 movies, respectively) using the same imaging conditions in a 200 kV Talos Arctica cryo-electron microscope (Thermo Fisher Scientific) equipped with a K3 direct detector and BioQuantum energy filter (Gatan). Micrographs were recorded with EPU software at a nominal magnification of X100,000,

corresponding to a pixel size of 0.830 Å. Each movie was dose-fractionated to 50 frames with a total exposure time of 3.4 s. The electron dose rate was 10.1 e⁻/pix/s (1.0 e⁻/Å²/frame), resulting in a total dose of 49.9 e⁻/Å²/frame. The defocus range was from −1.2 to −2.4 $\mu m$.

For the DIDS-bound form in nanodiscs, a dataset of 8,431 movies was collected using a 300 kV Titan Krios cryo-electron microscope (Thermo Fisher Scientific) equipped with a K3 direct detector and BioQuantum energy filter (Gatan). Micrographs were recorded with EPU software at a nominal magnification of X105,000, corresponding to a pixel size of 0.851 Å. Each movie was dose-fractionated to 50 frames with a total exposure time of 2.9 s. The electron dose rate was 14.7 e⁻/pix/s (1.2 e⁻/Å²/frame), resulting in a total dose of 60 e⁻/Å²/frame. The defocus range was from −0.8 to −2.0 $\mu m$. The cryo-EM data collection parameters for all datasets are summarized in Tables S1 and S2.

### Cryo-EM data processing

All data processing for INDY was performed in CryoSPARC v4.2.1 using a similar strategy (Punjani et al, 2017). Briefly, the raw images were subjected to patch-based motion correction and CTF estimation. Low-quality micrographs displaying low resolution, high defocus, a low figure of merit, and high astigmatism were excluded from further processing. A subset of images was used for automated particle picking using a blob picker, followed by extraction with a box size of 250 pixels. The particles were then subjected to 2D classification to remove junk particles, and the good classes were used for template-based particle picking from the complete set of micrographs. After multiple rounds of 2D classification, particles comprising the best 2D classes were selected to generate an ab initio model for 3D reconstruction, followed by 3D heterogeneous and homogeneous refinement. The particles in the major classes were further refined through global CTF refinement. The final 3D reconstruction was achieved using non-uniform refinement with C1 or C2 symmetry. The detailed cryo-EM data processing workflow for each dataset is presented in Figs S12, S13, S14, S15, S16, S17, S18, S19, S20, S21, S22, S23, S24, S25, S26, S27, S28, and S29.

### Model building and refinement

The previously reported crystal structure of *Vc*INDY in complex with citrate (PDB ID 5ULD) (Nie et al, 2017) was used as an initial model by fitting its individual domains into the map as rigid bodies in UCSF Chimera (Pettersen et al, 2004). The model was then refined using PHENIX real-space refinement (Adams et al, 2010). It was further improved by iterative rounds of manual model building in Coot (Emsley & Cowtan, 2004) and refinement in PHENIX. Regions showing poor electron density were modeled as polyalanine. Because of the poor substrate density of the map of citrate-bound INDY, the structure of *Vc*INDY was used as a reference to position the citrate molecule within the binding site. The final refined structure was validated using MolProbity (Chen et al, 2010). All figures were generated using PyMOL (Schrödinger, LLC, https://pymol.org/), UCSF Chimera, and Chimera X (Goddard et al, 2018). The

cryo-EM data collection parameters, refinement, and validation statistics for all datasets are summarized in Tables S1 and S2.

### Statistical analysis

Statistical analysis and graph production were performed using GraphPad Prism 10.0.0 software. One-way analysis of variance (ANOVA) followed by Dunnett's multiple comparison test, as indicated in figure legends, was employed for statistical comparisons. In figures, asterisks denote significance levels: ns indicates $P < 0.1234$, * indicates $P < 0.0332$, ** indicates $P < 0.0021$, *** indicates $P < 0.0002$, and **** indicates $P < 0.0001$. Experimental results are presented as the mean ± SEM.

## Data Availability

The atomic coordinates of the seven structures have been deposited in the Protein Data Bank with accession codes 8ZKW (apo, asymmetric, pH 6), 8ZL1 (apo, outward-open, pH 6), 8ZKZ (apo, asymmetric, pH 8), 8ZL6 (apo, inward-open, pH 8), 8ZL4 (citrate-bound, inward-partially open), 8ZL3 (DIDS-bound, outward-open), and 8ZL2 (DIDS-bound, asymmetric). The cryo-EM density maps have been deposited in the Electron Microscopy Data Bank with accession codes EMD-60210 (apo, asymmetric, pH 6), EMD-60215 (apo, outward-open, pH 6), EMD-60213 (apo, asymmetric, pH 8), EMD-60220 (apo, inward-open, pH 8), EMD-60218 (citrate-bound, inward-partially open), EMD-60217 (DIDS-bound, outward-open), and EMD-60216 (DIDS-bound, asymmetric).

## Supplementary Information

## Acknowledgements

We are grateful to Dr. Julian Gross for critical reading of the article. The use of cryo-EM facilities of the NEXUS consortium was supported by a National Research Foundation of Korea grant RS-2024-00440289. This work was supported by grants (NRF-2021M3A9I4022846, NRF-2022R1A2C1091278, RS-2024-00344154, and RS-2024-00440614) and postdoctoral fellowship (NRF-2021R1A6A3A01086747) from the National Research Foundation (NRF) funded by the Korean government (MSIT).

### Author Contributions

S Kim: conceptualization, validation, visualization, methodology, and writing—original draft, review, and editing.
JG Park: methodology.
SH Choi: methodology.
JW Kim: methodology.
MS Jin: conceptualization, supervision, funding acquisition, validation, project administration, and writing—original draft, review, and editing.

## Conflict of Interest Statement

The authors declare that they have no conflict of interest.

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
