## [Reviewer comments · Life Science Alliance]

Life Science Alliance

Cryo-EM structures reveal the H⁺/citrate symport mechanism of *Drosophila* INDY

Subin Kim, Jun Gyou Park, Seung Hun Choi, Ji Won Kim, and Mi Sun Jin

DOI: <https://doi.org/10.26508/lsa.202402992>

Corresponding author(s): *Mi Sun Jin, Gwangju Institute of Science and Technology*

Review Timeline:

Submission Date:	2024-08-12
Editorial Decision:	2024-09-20
Revision Received:	2025-01-07
Editorial Decision:	2025-01-09
Revision Received:	2025-01-10
Accepted:	2025-01-13

Scientific Editor: *Eric Sawey, PhD*

Transaction Report:

September 20, 2024

Re: Life Science Alliance manuscript #LSA-2024-02992-T

Prof. Mi Sun Jin
Gwangju Institute of Science and Technology
School of Life Sciences
Cheomdangwagi-ro, Buk-gu
Gwangju 61005
Korea, Republic of

Dear Dr. Jin,

Thank you for submitting your manuscript entitled "Cryo-EM structures reveal the cation-independent citrate transport mechanism of the *Drosophila* INDY" to Life Science Alliance. The manuscript was assessed by expert reviewers, whose comments are appended to this letter. We invite you to submit a revised manuscript addressing the Reviewer comments.

Thank you for this interesting contribution to Life Science Alliance. We are looking forward to receiving your revised manuscript.

Sincerely,

B. MANUSCRIPT ORGANIZATION AND FORMATTING:

Reviewer #1 (Comments to the Authors (Required)):

The authors report the structural and functional characterization of a sodium independent dicarboxylic acid transporter INDY from *Drosophila*. This protein belongs to DASS-E family and is therefore thought to be an exchanger for dicarboxylic acids, however the authors here curiously observe INDY to function as a symporter for citrate rather than an exchanger. INDY was solved in multiple conformations in the apo state at different pH as well in the presence of the native substrate, citrate as well as an inhibitor DIDS present preferentially in the outward open conformation. Together, the authors are able to describe the steps involved in substrate translocation as well as the structural basis for inhibition for the INDY family of transporters.

While the structural states for this transporter family have been described before, the additional partially occluded state with citrate bound and the role of the aromatic residue in the scaffold domain are nice additions.

In my opinion the most interesting aspect of this work is the loss of Na⁺-coupling and the observation that this protein can operate without a counter-substrate in proteoliposome assays. There are a number of elevator proteins that can use either Na⁺ or H⁺ as a driving ion and they are structurally similar. There is interesting paper looking at the evolution of H⁺ vs Na⁺ coupling in the glutamate transporters (<https://www.biorxiv.org/content/10.1101/2023.12.03.569786v2>). I would also recommend this recent review on ion-coupling in transporters (<https://www.nature.com/articles/s41586-024-07062-3>).

Based on the observation of pH dependence (both in terms of functional uptake and population distribution by cryo EM) and the fact that other elevator proteins can switch between either Na⁺ or H⁺ ions (depending on the organism) I would suggest that INDY is probably H⁺-coupled. This can be measured by using pH sensitive dyes and measuring activity in the presence of a H⁺-ionophore. These assays are straight forward to do and would improve the quality of the paper.

Reviewer #2 (Comments to the Authors (Required)):

The study by Kim et al. reports several structures and characterization of the INDY citrate transporter from *Drosophila*, a DASS-family transporter and prototype for understanding the family's physiological function. In a series of remarkable CryoEM studies, they report the first structures of this protein in multiple conformations. Additional transport assays enable them to characterize the protein's enzymatic mechanism.

The CryoEM experiments are well-designed and completed, and the results are significant and merit publication. However, details in the results are overly simplified or unexplored, which are critical to understanding the results and should be further described.

Critical concerns:

- The experimental results demonstrate the transporter has uniporter activity *in vitro*, and suggest the lipid environment may be the cause. This surprising finding should be further explored in the discussion, as it is a distinct reaction cycle from the canonical DASS exchangers and symporters and thereby has consequences for the transporter's mechanism, energetics, and physiology.
- The authors argue that the data in Fig 1A demonstrate the transporter's selectivity for citrate²⁻, but this arrangement also establishes a pH electrochemical gradient. Further, the authors note that apo DmINDY's conformation changes with pH. Both data suggest pH-dependent transport, and co-transported cations are required if the transporter is truly electroneutral, as previously reported. The authors should test transport with varied pHs (matched inside and outside the proteoliposome) to test substrate selectivity and driving force.
- The mutants designed to test the DIDS binding site might affect citrate transport. Therefore, citrate transport should be measured with and without DIDS, and the relative inhibition for each mutant should be used to quantify the effect of each amino acid change on activity.
- Absolute thermostability measurements for mutants could reflect simple changes in overall protein stability, and not necessarily changes in the mutant's ability to bind citrate. Therefore, the change in thermostability with and without substrate (ΔT_m) should be used to validate changes in substrate binding with each mutant.

Major concerns:

- While experimental density is shown for citrate and DIDS, the quality/significance of this density cannot be evaluated as the nearby protein is not shown. A dedicated figure showing the density of citrate or DIDS along with nearby protein at the same sigma level, should be added to the supplement.
- The authors mention UCSF DOCK failed to dock the citrate ligand. Are there other options?
- What occupies the DIDS site in the other reconstructions where the transport domain is in the outward-open position - is it empty or occupied by something (eg lipid)? What changes are seen in this regulatory site, or the canonical DASS binding site, upon DIDS binding?
- Did the authors note any changes in the structure with citrate or DIDS binding? This should be explicitly and quantitatively stated, including an RMSD.

Minor concerns:

- Line 203: The authors state that the control mutation S479A does not significantly affect transport. However, rather than no effect, the mutation has a statistically significant increase in transport over the wild type. This should be noted and briefly discussed.
- Slit width for the energy filter should be stated in the Methods
- Line 164: Authors state that INDY was 'mostly' in the inward occluded conformation. This should be quantified if possible. Were the authors able to isolate other states (even with modest resolution)?
- Occluded is probably the wrong term for the observed "inward-occluded" conformation, as citrate can leave in MD simulations. Instead, this appears to be one of two distinct inward-open conformations.
- Line 195-197: The authors argue the low efficiency of citrate transport at pH 7.5 is due to DmINDY having a relatively higher affinity for trivalent citrate, leading to inefficient substrate release. However, weak binding to citrate³⁻ could explain the same results. To support this hypothesis, authors should include affinity measurement.
- The exact conditions for the two NaCl measurements in Figure 5D are ambiguous. These two conditions should be clearly labeled for what is being tested.
- As the X-axis of Fig 1E is quantitative, this should be shown as a scatter plot, rather than a bar chart. The data should also be fit to extract an IC50 for DIDS.
- Figure 8 appears imprecise, as the reaction cycles don't match between the panels or the states observed in the CryoEM datasets. Admittedly, neatly illustrating non-cooperative dimers makes for a complex image. It might be better to simply show symmetric dimers and then explicitly state in the legend that the dimer is shown symmetrically for simplicity but the protomers are expected to operate independently.

Trivial concerns:

- Standard definition of cpm is 'counts per minute' not counters as shown on y axes of radioactive uptake assays.

Comments from Reviewers (Bold) and Author Responses**Reviewer #1 (Comments to the Authors (Required)):**

The authors report the structural and functional characterization of a sodium independent dicarboxylic acid transporter INDY from *Drosophila*. This protein belongs to DASS-E family and is therefore thought to be an exchanger for dicarboxylic acids, however the authors here curiously observe INDY to function as a symporter for citrate rather than an exchanger. INDY was solved in multiple conformations in the apo state at different pH as well in the presence of the native substrate, citrate as well as an inhibitor DIDS present preferentially in the outward open conformation. Together, the authors are able to describe the steps involved in substrate translocation as well as the structural basis for inhibition for the INDY family of transporters.

While the structural states for this transporter family have been described before, the additional partially occluded state with citrate bound and the role of the aromatic residue in the scaffold domain are nice additions.

In my opinion the most interesting aspect of this work is the loss of Na⁺-coupling and the observation that this protein can operate without a counter-substrate in proteoliposome assays. There are a number of elevator proteins that can use either Na⁺ or H⁺ as a driving ion and they are structurally similar. There is interesting paper looking at the evolution of H⁺ vs Na⁺ coupling in the glutamate transporters (<https://www.biorxiv.org/content/10.1101/2023.12.03.569786v2>). I would also recommend this recent review on ion-coupling in transporters (<https://www.nature.com/articles/s41586-024-07062-3>).

>> We thank the reviewer for highlighting this intriguing aspect of our work. We have reviewed the suggested references and cited them in the revised manuscript. Please see lines 103–106.

Based on the observation of pH dependence (both in terms of functional uptake and population distribution by cryo EM) and the fact that other elevator proteins can switch between either Na⁺ or H⁺ ions (depending on the organism) I would suggest that INDY is probably H⁺-coupled. This can be measured by using pH sensitive dyes and

measuring activity in the presence of a H⁺-ionophore. These assays are straight forward to do and would improve the quality of the paper.

>> We appreciate the reviewer's helpful comments and suggestions. In response, we performed liposome-based proton transport assays using the pH indicator pyranine. Notably, we observed that acidification of the lumen due to proton influx occurred only in the presence of citrate in the external buffer. Such acidification was not detected in the presence of succinate or DIDS. Valinomycin, a potassium ionophore, had no effect on INDY-mediated proton influx. These findings are discussed in the revised text (lines 102–116) and are presented in Figure 1F. To reflect these results, we have changed the title to "*Cryo-EM structures reveal the H⁺/citrate symport mechanism of Drosophila INDY*".

=====

Reviewer #2 (Comments to the Authors (Required)):

The study by Kim et al. reports several structures and characterization of the INDY citrate transporter from Drosophila, a DASS-family transporter and prototype for understanding the family's physiological function. In a series of remarkable CryoEM studies, they report the first structures of this protein in multiple conformations. Additional transport assays enable them to characterize the protein's enzymatic mechanism.

The CryoEM experiments are well-designed and completed, and the results are significant and merit publication. However, details in the results are overly simplified or unexplored, which are critical to understanding the results and should be further described.

Critical concerns:

The experimental results demonstrate the transporter has uniporter activity in vitro, and suggest the lipid environment may be the cause. This surprising finding should be further explored in the discussion, as it is a distinct reaction cycle from the canonical DASS exchangers and symporters and thereby has consequences for the transporter's mechanism, energetics, and physiology.

>> As clarified through the proton-based assays (see below), our results demonstrate that INDY functions as a H⁺/citrate symporter rather than a uniporter. Therefore, we have removed this section from the revised manuscript.

The authors argue that the data in Fig 1A demonstrate the transporter's selectivity for citrate²⁻, but this arrangement also establishes a pH electrochemical gradient. Further, the authors note that apo DmlNDY's conformation changes with pH. Both data suggest pH-dependent transport, and co-transported cations are required if the transporter is truly electroneutral, as previously reported. The authors should test transport with varied pHs (matched inside and outside the proteoliposome) to test substrate selectivity and driving force.

>> To address this issue, we performed a number of experiments to optimize the liposome system and pH indicators (pyranine and BCECF). Using pyranine, we confirmed that INDY indeed facilitates H⁺/citrate co-transport into liposomes. Specifically, we observed that lumen acidification due to proton influx occurred only in the presence of citrate in the external buffer. Such acidification was not detected in the presence of succinate or DIDS. Valinomycin, a potassium ionophore, had no effect on INDY-mediated proton influx, suggesting that transport is indeed electroneutral. These findings are discussed in the revised manuscript (lines 102–116) and are presented in Figure 1F.

>> We also attempted various combinations of pH and citrate gradients across the liposomes; however, accurate ratiometric or quantitative comparisons between datasets proved unreliable, presumably due to several experimental limitations. For example, variation in charged citrate concentrations can introduce inconsistencies in pH measurements. Additionally, achieving uniform encapsulation of pyranine within liposomes during preparation is inherently challenging and can result in sample variability. Consequently, the effects of pH and substrate gradients could not be conclusively analyzed in this study.

The mutants designed to test the DIDS binding site might affect citrate transport. Therefore, citrate transport should be measured with and without DIDS, and the relative inhibition for each mutant should be used to quantify the effect of each amino acid change on activity.

>> As recommended, we performed additional experiments to measure citrate transport in the designed mutants with and without DIDS, and quantified the relative % inhibition caused by DIDS. The results are presented in Figure 7D and Figure S10E, and discussed in lines 304–308 in the revised manuscript.

Absolute thermostability measurements for mutants could reflect simple changes in overall protein stability, and not necessarily changes in the mutant's ability to bind citrate. Therefore, the change in thermostability with and without substrate (ΔT_m) should be used to validate changes in substrate binding with each mutant.

>> Following this advice, we have revised Figures 5F and G to show citrate binding-induced changes in thermostability (ΔT_m) for each mutant.

Major concerns:

While experimental density is shown for citrate and DIDS, the quality/significance of this density cannot be evaluated as the nearby protein is not shown. A dedicated figure showing the density of citrate or DIDS along with nearby protein at the same sigma level, should be added to the supplement.

>> Figures 5A and 7C have been modified to show the electron density for citrate and DIDS, along with the nearby key residues at the same 3.5 σ level.

The authors mention UCSF DOCK failed to dock the citrate ligand. Are there other options?

>> We also explored alternative docking programs including AutoDock Vina and CB-Dock. However, they similarly failed to position citrate in the binding site in a way that aligned with our mutagenesis analysis results.

What occupies the DIDS site in the other reconstructions where the transport domain is in the outward-open position - is it empty or occupied by something (eg lipid)?

>> In the apo outward-open maps, the DIDS-binding site was found to be empty.

What changes are seen in this regulatory site, or the canonical DASS binding site, upon DIDS binding?

>> No significant structural changes were observed in the substrate-binding site upon DIDS binding, as shown in Figure S10D.

Did the authors note any changes in the structure with citrate or DIDS binding? This should be explicitly and quantitatively stated, including an RMSD.

>> In the revised manuscript, we have included an analysis of the structural changes associated with citrate and DIDS binding (see lines 172–174, 295–296, and Figures S6C and S10D).

Minor concerns:

Line 203: The authors state that the control mutation S479A does not significantly affect transport. However, rather than no effect, the mutation has a statistically significant increase in transport over the wild type. This should be noted and briefly discussed.

>> One possible explanation for the increased transport activity observed for the S479A mutant could be that the mutation alleviates steric hindrance or alters the local hydrogen bonding network within the binding pocket, thereby facilitating substrate binding and/or translocation. This has been noted in the revised text (lines 212–216).

Slit width for the energy filter should be stated in the Methods

>> We have now included the slit width for the energy filter in the Methods section (see line 545).

Line 164: Authors state that INDY was 'mostly' in the inward occluded conformation. This should be quantified if possible. Were the authors able to isolate other states (even with modest resolution)?

>> In the revised manuscript, we have now quantified the occupancy of the inward-occluded conformation in our data. We confirmed that other classes also exhibited a nearly identical inward-occluded conformation, but were excluded from the final model to enhance the resolution (see lines 175–177).

Occluded is probably the wrong term for the observed "inward-occluded" conformation, as citrate can leave in MD simulations. Instead, this appears to be one of two distinct inward-open conformations.

>> We have changed "inward-occluded" to "inward-partially open" throughout the manuscript.

Line 195-197: The authors argue the low efficiency of citrate transport at pH 7.5 is due to DmINDY having a relatively higher affinity for trivalent citrate, leading to inefficient substrate release. However, weak binding to citrate³⁻ could explain the same results. To support this hypothesis, authors should include affinity measurement.

>> We attempted to measure and compare the citrate affinity of INDY at different pH values

using microscale thermophoresis (MST). Despite our extensive efforts, it was challenging to obtain accurate K_d values, possibly due to protein aggregation during the measurement or adsorption to the capillary surface. However, we fully understand the reviewer's concern and have therefore deleted this sentence from the manuscript.

The exact conditions for the two NaCl measurements in Figure 5D are ambiguous. These two conditions should be clearly labeled for what is being tested.

>> If you are referring to Figure 1D, we have revised the figure and its corresponding legend to clearly specify the exact conditions for the liposome assays.

As the X-axis of Fig 1E is quantitative, this should be shown as a scatter plot, rather than a bar chart. The data should also be fit to extract an IC50 for DIDS.

>> As suggested, we have revised Figure 1E to present the data as a scatter plot, and have calculated the IC₅₀ value. Please refer to lines 99–100.

Figure 8 appears imprecise, as the reaction cycles don't match between the panels or the states observed in the CryoEM datasets. Admittedly, neatly illustrating non-cooperative dimers makes for a complex image. It might be better to simply show symmetric dimers and then explicitly state in the legend that the dimer is shown symmetrically for simplicity but the protomers are expected to operate independently.

>> We agree that the original Figure 8 may have caused confusion. In the revised manuscript, we have simplified the model of the transport cycle by depicting only one protomer for clarity. We have also added a note in the figure legend stating that only one protomer per INDY dimer is shown for simplicity, and that protomers are believed to operate independently.

Trivial concerns:

Standard definition of cpm is 'counts per minute' not counters as shown on y axes of radioactive uptake assays.

>> Thank you for pointing this out. We have corrected this mistake in the revised manuscript.

January 9, 2025

RE: Life Science Alliance Manuscript #LSA-2024-02992-TR

Prof. Mi Sun Jin
Gwangju Institute of Science and Technology
School of Life Sciences
Cheomdangwagi-ro, Buk-gu
Gwangju 61005
Korea, Republic of South Korea

Dear Dr. Jin,

Thank you for submitting your revised manuscript entitled "Cryo-EM structures reveal the H⁺/citrate symport mechanism of *Drosophila* INDY". We would be happy to publish your paper in Life Science Alliance pending final revisions necessary to meet our formatting guidelines.

- please be sure that the authorship listing and order is correct
- please upload your Tables in editable .doc or excel format
- please add a Running Title and a Summary Blurb/Alternate Abstract in our system in our system
- please add Keywords for your manuscript to our system
- please add the Twitter handle of your host institute/organization as well as your own or/and one of the authors in our system
- please add your main, supplementary figure, and table legends to the main manuscript text after the references section
- please remove legends from the supplementary figures
- please revise the legend for Figure S7 so that the panels are introduced in order
- please be sure to add call-outs for all panels in all supplementary figures
- you may want to consider uploading Figure 8 as a Graphical Abstract rather than as a figure, but this is up to you

LSA now encourages authors to provide a 30-60 second video where the study is briefly explained. We will use these videos on social media to promote the published paper and the presenting author (for examples, see <https://docs.google.com/document/d/1-UWCfbE4pGcDdcgzcmiuJl2XMBJnxKYeqRvLLrLS08s/edit?usp=sharing>). Corresponding or first-authors are welcome to submit the video. Please submit only one video per manuscript. The video can be emailed to contact@life-science-alliance.org

A. FINAL FILES:

B. MANUSCRIPT ORGANIZATION AND FORMATTING:

Sincerely,

January 13, 2025

RE: Life Science Alliance Manuscript #LSA-2024-02992-TRR

Prof. Mi Sun Jin
Gwangju Institute of Science and Technology
School of Life Sciences
Cheomdangwagi-ro, Buk-gu
Gwangju 61005
Korea, Republic of South Korea

Dear Dr. Jin,

Thank you for submitting your Research Article entitled "Cryo-EM structures reveal the H⁺/citrate symport mechanism of *Drosophila* INDY". It is a pleasure to let you know that your manuscript is now accepted for publication in Life Science Alliance. Congratulations on this interesting work.

DISTRIBUTION OF MATERIALS:

Again, congratulations on a very nice paper. I hope you found the review process to be constructive and are pleased with how the manuscript was handled editorially. We look forward to future exciting submissions from your lab.

Sincerely,
